# Histological Correlation between Tonsillar and Glomerular Lesions in Patients with IgA Nephropathy Justifying Tonsillectomy: A Retrospective Cohort Study

**DOI:** 10.3390/ijms25105298

**Published:** 2024-05-13

**Authors:** Kensuke Joh, Hiroyuki Ueda, Kan Katayama, Hiroshi Kitamura, Kenichi Watanabe, Osamu Hotta

**Affiliations:** 1Department of Pathology, The Jikei University School of Medicine, Tokyo 105-8461, Japan; 2Division of Nephrology and Hypertension, Department of Internal Medicine, The Jikei University School of Medicine, Tokyo 105-8461, Japan; uehiroriheu@gmail.com; 3Department of Cardiology and Nephrology, Mie University Graduate School of Medicine, Tsu 514-8507, Japan; katayamk@med.mie-u.ac.jp; 4Division of Research Center, Japan Community Health Organization (JCHO) Sendai, Sendai 981-3281, Japan; kitamura-hiroshi@sendai.jcho.go.jp; 5Department of Otolaryngology, Head and Neck Surgery, Tohoku University School of Medicine, Sendai, Sendai 980-8575, Japan; kenw@orl.med.tohoku.ac.jp; 6Department of Otolaryngology, Tohoku Rosai Hospital, Sendai 981-8563, Japan; 7Division of Internal Medicine, Hotta Osamu Clinic (HOC), Sendai 984-0013, Japan; hottao@remus.dti.ne.jp

**Keywords:** IgA nephropathy, tonsillitis, glomerulonephritis, tonsillectomy, steroid pulse therapy, pathology

## Abstract

Tonsillectomy with steroid pulse therapy (SPT) has been established as an effective treatment for immunoglobulin A nephropathy (IgAN) in Japan. However, the underlying mechanisms supporting tonsillectomy remain unclear. This study assessed palatine tonsils from 77 patients with IgAN, including 14 and 63 who received SPT before and after tonsillectomy, respectively. Tonsils from 21 patients with chronic tonsillitis were analyzed as controls. Specific tonsillar lesions were confirmed in patients with IgAN, correlating with active or chronic renal glomerular lesions and SPT. T-nodule and involution of lymphoepithelial symbiosis scores in tonsils correlated with the incidence of active crescents and segmental sclerosis in the glomeruli, respectively. The study revealed an essential role of the tonsil–glomerular axis in early active and late chronic phases. Moreover, the SPT-preceding group demonstrated no changes in the T-nodule score, which correlated with active crescent formation, but exhibited a considerable shrinkage of lymphatic follicles that produced aberrant IgA1. The study underscores the involvement of innate and cellular immunity in IgAN and advocates for tonsillectomy as a necessary treatment alongside SPT for IgAN, based on a stepwise process.

## 1. Introduction

The therapeutic regimens for immunoglobulin A nephropathy (IgAN) administered in most parts of the world include renin–angiotensin–aldosterone system inhibitors (RAASi) and other novel agents, such as sodium–glucose co-transporter 2 inhibitors (SGLT2i), which have been shown to be efficient. However, tonsillectomy is not recommended worldwide [1]. In Japan, however, tonsillectomy with steroid pulse therapy (SPT) has been recognized as an effective treatment for IgAN in recent years [2,3,4,5,6,7]. A study found that 137 out of 188 hospitals (66.2%) had begun to perform tonsillectomy with SPT in the period from 2004 to 2008 [8]. While several reviews advocate for a shift from “slowing the progression of nephropathy” to “remission of nephropathy”, few studies have provided histological evidence concerning the correlation between the tonsil and kidneys, suggesting the necessity of tonsillectomy in addition to SPT [4,9,10,11,12,13,14].

In our facility, we included patients with a high disease progression detected in renal biopsy who had undergone SPT before tonsillectomy. Thus, it was possible that the tonsils were histologically compared between the SPT- and tonsillectomy-preceding groups. Therefore, our study was conducted to confirm the following points: the presence of any specific tonsillar lesions in patients with IgAN, the relationship of these lesions with active or chronic renal glomerular lesions, and the association of SPT with these lesions. If SPT does not affect tonsillar changes, which are associated with renal lesions, we hypothesized that removing tonsils, which are implicated in the cause of renal lesions, could yield more effective results aiming at IgAN remission but not merely inhibiting IgAN progression. Therefore, we aimed to find any evidence that tonsillectomy is a necessary choice of treatment instead of only SPT for IgAN based on this stepwise process.

## 2. Results

### 2.1. Pathological Differences between TON-A and TON-C

We initially aimed to identify gross morphological differences between tonsil tissues of patients with IgAN and healthy individuals. To achieve this, we assessed the weight, layered cell structure, and T-nodule scores for each group. The average weight of TON-A was 3.1 ± 1.5 g and demonstrated a significantly lower value than that of TON-C, which was 5.8 ± 2.8 g (*p* < 0.05). TON-C exhibits regular round-shaped follicles with well-developed germinal centers with relatively thin mantle zones encircling relatively narrow marginal zones of uniform width (Figure 1a). The areas, which were positive for CD4, were located primarily in the mantle zone and germinal centers (Figure 1b). CD8 was detected predominantly in the former but not in the latter (Figure 1c). The area of the dorsal side of the lymphoid follicle, which was positive for CD4 and CD8, was limited and not prominent. Conversely, TON-A demonstrated shrinking of germinal centers, while the mantle zone was shaped as an elliptical bulge toward the crypts. The germinal center is positioned in the deep regions of the latter (Figure 1d). The area, which demonstrated CD4 staining, was prominent on the dorsal side of the lymphoid follicle, as the opposite side of the crypt was also positive for CD8. CD4-positive T-lymphocytes were abundant in mantle zones and more numerous in extrafollicular areas and germinal centers, whereas CD 8-positive T-cells were mainly detected in the mantle zone and were more prominent in an extrafollicular area but not in a germinal center. In these areas, the CD4-positive lymphocytes were more frequently detected than CD8-positive cells in both TON-A and TON-C (Figure 1e,f).

CD 3-positive T-cells in Ton-A were mainly detected in T-nodules and slightly positive in mantle zones and the dark area in germinal centers (Figure 2a). CD20-positive B cells were found in the mantle zone but not in the T-nodule area (Figure 2b). An assembly of HLA-DP, DQ, and DR-positive cells formed the cluster in a T-nodule. In the center of the CD3-positive T-nodule area, an assembly of HLA-DP, DQ, and DR-positive cells was detected (arrows, Figure 2c), whereas a lymphatic follicle that consisted of a germinal center and mantle zone was also positive for HLA- DP, DQ, and DR. CD208-positive mature dendritic cells (DCs) belonging to the lysosome-associated membrane glycoprotein family (LAMP)-3 (DC-LAMP-3) were detected in the area of HLA-DP, DQ, and DR-positive dendritic cells (arrows, Figure 2d) [15]. 

HEVs were stained with anti-CD 34 antibodies and were distributed in the areas of T-nodules (Appendix A). Lymphatic vessels were stained with anti-D2-40 antibodies and were concentrated in the same areas of T-nodules (Appendix A). 

IgA- and IgG-bearing plasma cells were distributed around the mantle zone of lymphatic follicles beneath the tonsillar crypt epithelium (Appendix A). Overall, tonsillar slices of 15 and 6 patients of TON-A and TON-C were stained with anti-IgG and anti-IgA antibodies, respectively. Five locations (0.8 mm^2^ per location) for estimating the frequencies of IgA- and IgG-bearing plasma cells were counted by an image analyzer (NIS Elements D 3.1) and the ratios of IgA-/IgG-bearing plasma cells of TON-A and TON-C were compared.

In TON-A, the numbers of IgA- and IgG-bearing plasma cells were reported as 121 ± 54/0.8 and 182 ± 74/0.8 mm^2^, respectively, while in TON-C, 83 ± 38/0.8 and 155 ± 38/0.8 mm^2^ of IgA- and IgG-bearing cells were detected, respectively. The IgA/IgG ratio was 0.71 ± 32 in TON-A, while it was 0.58 ± 38 in TON-C (*p* < 0.05). Therefore, TON-A exhibited a higher ratio of IgA- to IgG-bearing plasma cells than TON-C (*p* < 0.01) (Appendix A).

The comparison of the T-nodule and ILES scores between TON-A and TON-C quantitatively revealed T-nodules of 3.9 + −0.7 in TON-A and 3.0 + −1.0 in TON-C, with a significant difference (*p* < 0.01) (Figure 3a). The ILES scores were 1.9 + −1.0 in TON-A and 1.0 + −0.9 in TON-C, with a significant difference (*p* < 0.01) (Figure 3b). Both indices showed significantly higher values in TON-A compared to TON-C.

These findings suggest that the T-nodule and ILES scores were used to investigate whether these tonsillar lesions correlated with renal lesions.

The statistical relationship between lesions of TON-A and renal lesions, including mesangial hypercellularity, intraglomerular hypercellularity, active crescentic lesions as either cellular or fibrocellular crescents, focal segmental, global sclerotic lesions among the total number of glomeruli, and tubular atrophy/interstitial fibrosis as 10% of the total number of increments, was investigated. Multivariate linear regression analysis revealed that only active crescent and segmental sclerosis correlated significantly with T-nodule and ILES scores, respectively (Table 1 and Table 2) (*p* < 0.05). Therefore, the tonsillar–glomerular axis was strongly associated with early active and chronic phases.

### 2.2. T-Nodule Scores in the SPT- and Tonsillectomy-Preceding TON-A and TON-C Groups

The study was conducted to confirm the association of SPT with the T-nodule score in the TON-A and TON-C groups. The average T-nodule scores in the 63 cases receiving SPT after tonsillectomy (tonsillectomy-preceding group), 14 patients undergoing SPT before tonsillectomy (SPT-preceding group) of TON-A, and 21 patients of TON-C were 3.9 + −0.7, 3.9 + −0.9, and 3.2 + −1.0, respectively. The average T-nodule score demonstrated no difference between the tonsillectomy-preceding and SPT-preceding groups. However, the T-nodule scores in the TON-C group were significantly lower than those of tonsillectomy-preceding and SPT-preceding TON-A groups (*p* < 0.05) (Figure 4a). Crescents were determined in 13 (20.6%) out of 63 patients in the tonsillectomy-preceding group and 5 (35.7%) out of 14 patients in the SPT-preceding group. Moreover, the average number of crescents was significantly higher in the SPT-preceding group (2.8 + −5.1) than in the tonsillectomy-preceding group (1.0 + −2.1) (*p* < 0.05) (Figure 4b). Therefore, we compared the average value of the T-nodule score between the 50 and 9 patients of the tonsillectomy-preceding and SPT-preceding groups without crescents, respectively. No difference was found between the two groups, respectively (3.8 + −0.6 vs. 3.8 + −0.9) (Figure 4c).

### 2.3. Staining of Tonsil Tissue from the SPT-Preceding and Tonsillectomy-Preceding Groups with Anti-HLA-DP, DQ, and DR Antibodies

The lymph follicles comprised of a germinal center and mantle zone were preserved with a marked number of T-nodules in one patient (24-year-old female) in the tonsillectomy-preceding TON-A group, whereas lymphatic follicles were severely destroyed, but the frequency of T-nodules was maintained in another patient (38-year-old male) in the SPT-preceding group who underwent three SPT courses 17 days before tonsillectomy (Figure 5a,b).

### 2.4. ILES Scores in the SPT-Preceding and Tonsillectomy-Preceding TON-A and TON-C Groups

The study was conducted to confirm the association of SPT with the ILES score in the TON-A and TON-C groups. The ILES score was significantly higher in the SPT-preceding group (2.8 ± 1.3) than in the tonsillectomy-preceding group (1.7 ± 0.7) in TON-A and TON-C (1.2 ± 0.9) (*p* < 0.01). This result implies that SPT has an effect on the enhancement of ILES scores (Figure 6).

### 2.5. Correlation between ILES and T-Nodule Scores in the SPT- and Tonsillectomy-Preceding TON-A and TON-C Groups

The study was conducted to analyze the correlation between the ILES and T-nodule score under the effect of SPT. A significant negative correlation was found between ILES and T-nodule scores in 14 patients in the SPT-preceding group (*p* < 0.05, 95% confidence interval [CI]: −0.71 to −0.03), but not in 63 patients in the tonsillectomy-preceding group (*p* > 0.05, 95% CI: −0.18 to 0.29), and in 21 patients in the TON-C group (*p* > 0.05, 95% CI: −0.04 to 0.91). This result demonstrated the negative feedback of ILES to the T-nodule in TON-A especially in the SPT-preceding group, where rapid progression was supposed to have occurred (Figure 7).

## 3. Discussion

We aimed to provide any evidence supporting tonsillectomy as a necessary treatment alongside SPT for IgAN, based on the following stepwise process.

First, the germinal center of lymphatic follicles was developed poorly with the extensive spindle-shaped marginal zone for any specific lesions in TON-A. The background of the shrinking lymph follicles shows the enlarged interfollicular T-cell zone defined as a T-nodule. These areas demonstrated the clusters of HLA-DP, DQ, and DR-positive dendric cells in the center, where CD208-positive mature dendritic cells (DC LAMP-3) were found. In contrast, TON-C exhibited regularly round-shaped follicles with enlarged germinal centers and poorly developed T-nodules. Kawaguchi and Takechi previously reported that the enlarged T-nodule structures in the interfollicular areas of tonsils were characteristic of patients with IgAN [16,17]. The enlarged T-nodules promoting active T-cell recruitment and proliferation, where CD208 is expressed exclusively on mature DCs, are an integral part of the processing and presentation of antigens on naive T-cells during the innate immune response. The naive T-cells flow from HEV (Appendix A) and come into contact with CD208-positive mature dendritic cells transforming into memory T-cells [15], which will be sent into systemic circulation via efferent lymph vessels (Appendix A). Antigen-presenting cells, such as membranous epithelial (M cells) and dendritic cells, are distributed in the crypt epithelium in a lymphoepithelial symbiosis site and may serve as the onset of antigen targeting in tonsils [18]. In contrast, the ILES score was significantly increased in TON-A compared with that of TON-C. Sato et al. also revealed that the non-reticulated crypt epithelium was frequently detected and consisted of >50% of the total crypt epithelium in TON-A, whereas the non-reticulated keratinizing area was composed of <7% of the total crypt epithelium in TON-C [19]. These results indicated that the increased ILES score showing the keratinization of the crypt epithelium inhibited the antigen presentation of dendritic cells in the crypt epithelium resulting from the involution of germinal centers of lymphatic follicles in TON-A [20].

Second, the T-nodule and ILES scores significantly correlated with the frequency of active crescents and segmental sclerosis, respectively. These outcomes conformed to those reported by Takechi et al., which verified that only CD208(+) DCs out of the distinct five subtypes of DCs, including CD303, CD1c, CD209, CD208, and CD1a, were significantly increased in TON-A compared with that of TON-C and were related to the proportion of crescentic glomeruli using immunohistochemistry and flow cytometry [17,21]. However, little is known about the mechanism involving the relationship between proliferated CD208-positive DCs in T-nodules in tonsils and an increase in active crescents in kidneys. The differentiated memory T-cells, specifically CD8/CX3CR1 double-positive cells, reach renal glomeruli and induce fractalkine as a ligand of CX3CR1 on glomerular endothelial cells, causing an influx of inflammatory cells in glomerular capillaries because fractalkine is considered the only ligand for the CX3CR1 receptor [12,13,21,22,23,24]. Recruitment of CXCR3+ T-cells in the kidney of adult patients with IgA vasculitis correlates with the involvement of the CX3CR1–fractalkine axis in the exacerbation of gross hematuria [25]. Conversely, humoral immunity plays a role in IgAN development. Class-switching from IgG-producing plasma cells to IgA occurs around lymphoid follicles under the influence of BAFF (B cell activating factor belonging to the tumor necrosis factor family) (Appendix A) [26,27,28,29]. Hence, the aberrant IgA1 production in the tonsils leads to its deposition on the glomerular mesangium. However, the direct impact of IgA1 deposition in the mesangium on the induction inflammatory cell influx promoting crescent formation is less likely, as marked IgA1 deposition in the mesangium often occurs without inflammation [12]. Our study confirms the role of cellular immunity in the tonsillar–glomerular relationship. In addition to T-nodule involvement in crescent formation, ILES scores significantly correlate with the frequency of segmental sclerosis in TON-A. Sato et al. revealed that the amount of glomerular damage was positively connected with the percentage of ILES [19]. An increase in the ILES score indicates inhibition of further antigen presentation on naive T-cells via dendric cells due to keratinization of the tonsillar crypt epithelium. A decrease in reactive response in the tonsils implies hindered active crescent formation and favors gradual progression alongside glomerular segmental sclerosis. Thus, an interaction between tonsils and glomeruli within the tonsillar–glomerular axis is strongly indicated in early active and chronic phases. The significant negative correlation between ILES and T-nodule scores in the SPT group reveals a negative feedback loop of ILES to the T-nodule (Figure 7).

Third, we confirmed that the T-nodule score did not significantly differ between the SPT-preceding and tonsillectomy-preceding groups in the whole cohort and even in the cohort of patients without crescent formation. Moreover, the T-nodule score remained stable, whereas lymph follicles were significantly sensitive to involution in the SPT-preceding group, as shown in Figure 5. This result reveals the preserved correlation between the T-nodule score and crescent formation even in the subgroup of the SPT-preceding group. Therefore, rationale for tonsillectomy in treating patients with IgAN was provided to ameliorate crescent formation as glomerular vasculitis. A consequent question is how SPT affects the prognosis of patients with IgAN besides tonsillectomy. SPT alone is generally effective in preventing glomerular tuft necrosis and cellular crescent development [30,31]. Furthermore, SPT after performing tonsillectomy is necessary because memory T-cells, which are already distributed in systemic circulation, can undergo apoptosis induced by SPT. Hence, SPT plays another important role in confirming the effect of tonsillectomy on making prognoses [32]. Therefore, studies have investigated the influence of tonsillectomy alone without SPT on renal prognosis by focusing solely on surgical intervention, where the outcomes significantly vary [33]. Since IgAN is regarded as a chronic progressive disease, the main focus of therapy using RAASi and SGLT2i has been on ameliorating progression and extending the period before end-stage renal disease by reducing proteinuria but not improving hematuria. However, clinically, IgAN begins with hematuria against the background of glomerular vasculitis and develops gradually into the proteinuria-dominant phase as a result of transition from the histopathological features of glomerular vasculitis to segmental sclerosis or adhesion, with persisting concurrent hematuria. Therefore, initial treatment is desirable during the period of acute active lesions exhibiting dominant hematuria. When proteinuria develops, its treatment can potentially alleviate renal functional decline but has no promising chance of remission of IgAN [12]. Moreover, Ohya et al. revealed that tonsillectomy conjugated with SPT lowered the relapse rate in patients with IgAN [34]. As annual urine laboratory analyses are widely conducted in Japan, IgAN is likely to be detected at an earlier stage than in other countries. Thus, suppressing glomerular vasculitis at an early stage by conducting tonsillectomy besides SPT is currently recommended as a goal for the shift from “inhibiting IgAN progression” to “IgAN remission”.

## 4. Materials and Methods

### 4.1. Patients

Eligible patients with IgAN, who were diagnosed via renal biopsy, were enrolled in the Japan Community Health Organization (JCHO) Sendai Hospital between 2010 and 2013. This study enrolled 77 patients (TON-A) (age: 14–69 years, median: 39 years), including 14 and 63 patients who received SPT before (SPT-preceding group) and after tonsillectomy (tonsillectomy-preceding group), respectively. SPT was conducted after tonsillectomy in the majority of cases; however, SPT was started preoperatively at the discretion of the attending physician of the patients with supposed rapid progression based either on the findings of renal biopsy or urinalysis or who preferred SPT due to the convenience because an interval of 2–3 months is required from kidney biopsy to tonsillectomy. Of the 14 patients in the SPT-preceding group, 1, 11, 2, and 3 underwent 4, 3, 2, and 1 SPT courses, respectively. The period from the start of SPT to tonsillectomy was 10–82 days.

The control group consisted of 21 patients with chronic tonsillitis (TON-C) in the recurrent form (6–71 years old, median: 29 years old). TON-A was identified in the patients who were diagnosed with IgAN by renal biopsy. Patients who did not have microhematuria, which is a characteristic of IgAN, were assigned to the TON-C group because few of them underwent renal biopsy. Patients with other systemic diseases that involve glomerular IgA deposition, such as IgA vasculitis, liver cirrhosis, and systemic lupus erythematosus, were excluded.

### 4.2. Study Approval

This retrospective cohort pathological study complied with the ethical requirements of JCHO Sendai and obtained informed consent from all the patients. The patients’ records and tissue samples were retrospectively assessed, suggesting general consent for research use but not specific consent for this study. The Tohoku University School of Medicine’s Institutional Review Board on Human Research (http://www.hosp.tohoku.ac.jp/privacy.html accessed on 25 January 2016) and the primary ethics committee for our research (no. 2015-1-679) approved the protocol, and this was declared on the following website: http://www.bureau.tohoku.ac.jp/kokai/disclosure/index.html accessed on 25 January 2016. Additionally, our study was performed according to the Declaration of Helsinki guidelines for good clinical practice.

### 4.3. Immunohistochemical Staining of Palatine Tonsils

After tonsillectomy, the tonsil tissues were immersed in 10% formaldehyde in phosphate-buffered saline and then in paraffin. The samples were sectioned for immunohistochemistry. The obtained slices were stained by VENTANA Bench Mark ULTRA with cell conditioning 1 (CC1) buffer for 36 min with the following primary antibodies: anti-HLA-DP, DQ, and DR mouse monoclonal, anti-CD208 (LAMP3) rabbit polyclonal, anti-CD3, anti-CD4, anti-CD8, and anti-CD20 (L26) mouse monoclonal, anti-CD34 for high epithelial venule (HEV), anti-D2-40 for lymph vessels, anti-cytokeratin (AE1/AE3), and anti-IgG and IgA antibodies for plasma cells. VENTANA Bench Mark ULTRA (Roche) was used to stain the frozen sections of all renal biopsy samples for IgG, IgA, IgM, C3, and C1q (the precise staining conditions are provided in the Appendix A). The diagnosis of IgAN was based on the immunofluorescent detection of glomerular IgA deposition.

### 4.4. Quantitative Analysis of Palatine Tonsils

Tonsils are composed of four tissue areas playing specific roles in immunity, namely the reticular crypt epithelium, mantle zones of lymphoid follicles, follicular germinal centers, and extrafollicular areas [18,35]. Measurements were conducted from one section in each case, analyzing at least five locations of the crypts of a palatine tonsil within the same tissue. The area of each part was assessed in serial slices.

### 4.5. T-Nodule Score

Anti-HLA-DP, DQ, and DR were used to evaluate the assembly of positive cells in the center of the interfollicular T-cell area, known as the T-nodule [18,35]. The number of cells that are positive for HLA-DP, DQ, and DR was counted in a 2.0 mm^2^ area and denoted as the T-nodule score for the quantitative analysis of the T-nodule (arrowheads, Figure 8a).

### 4.6. Involution of Lymphoepithelial Symbiosis (ILES) Score

The tonsillar crypts have a characteristic network-like structure called lymphoepithelial symbiosis, where the crypt epithelium and lymphocytes coexist [18]. The former, as detected by anti-cytokeratin (AE1/AE3) antibodies, demonstrated an extensive loose reticular pattern of lymphoepithelial symbiosis. ILES revealing the loss area of the reticular structure of the crypt epithelium transforming into keratotic epithelium was measured. The ILES score was identified as the length of the area of loose reticular patterns as the keratotic squamous epithelium in the crypt, and was measured using a curvimeter along the contour of the crypt epithelium (dotted line, Figure 8b). The percentage (%) of the length of the transformed keratotic epithelium out of the entire length of the crypt epithelium was estimated and showed 20% increments graded from 1 to 5 and expressed as the ILES score.

### 4.7. Pathological Analysis of Kidneys

Kidney tissue specimens were obtained via percutaneous needle biopsy to confirm IgAN diagnosis. The tissues were embedded in paraffin, cut into 3 µm sections, and stained with hematoxylin–eosin, periodic acid–Schiff, Masson’s trichrome, and periodic acid silver methenamine. The frozen sections from the kidney tissue underwent immunofluorescence for IgG, IgA, IgM, C3, and C1q. The diagnosis of IgAN was based on the presence of co- and dominant glomerular IgA deposition by immunofluorescence and that of electron-dense paramesangial deposits detected by electron microscopy. All pathological findings were performed under the Oxford classification [36]. We defined the percentage of glomeruli with mesangial hypercellularity, intraglomerular hypercellularity, active crescentic lesions as either cellular or fibrocellular crescents, and focal segmental and global sclerotic lesions among the total number of these renal structures. Tubular atrophy/interstitial fibrosis was detected in 10% of increments. Two blinded observers (KJ and HK) scored the pathological variables of renal biopsies. The same procedure was repeated to achieve a consensus in the case of disagreement.

### 4.8. Statistical Analyses

The Statistical Package for the Social Sciences version 24 (IBM Corp., Armonk, NY, USA) was used for statistical analysis. Mean and standard deviation were calculated for variables with normal distribution, while median and interquartile range (IQR) were used to describe non-normally distributed data. The Wilcoxon–Mann–Whitney two-sample rank sum test was applied to comparatively analyze continuous non-parametric data between the two groups. A *p*-value of <0.05 was established as the threshold for statistical significance.

We performed multivariate linear regression analysis to explore the association between T-nodule or ILES scores and glomerular lesions including mesangial hypercellularity, active (cellular or fibrocellular) crescents, fibrous crescents, global sclerosis, segmental sclerosis, and interstitial fibrosis/tubular atrophy.

## 5. Conclusions

The present study confirmed the strong association between palatine tonsillar and glomerular lesions in patients with IgAN. The T-nodule and ILES scores correlated with the frequency of active crescents and segmental sclerosis, respectively. The correlation between the T-nodule score and crescent formation rate was unaffected even in the subgroups of the SPT-preceding group, whereas the regression of lymphatic follicles was detected. The study revealed an essential role of the tonsil–glomerular axis in the early active as well as in the late chronic phases. Moreover, the SPT-preceding group demonstrated no change in the T-nodule score, which is associated with active crescent formation. These results support the rationale for tonsillectomy alongside SPT as a therapy targeting innate and cellular immunity in IgAN.

## Figures and Tables

**Figure 1 ijms-25-05298-f001:**
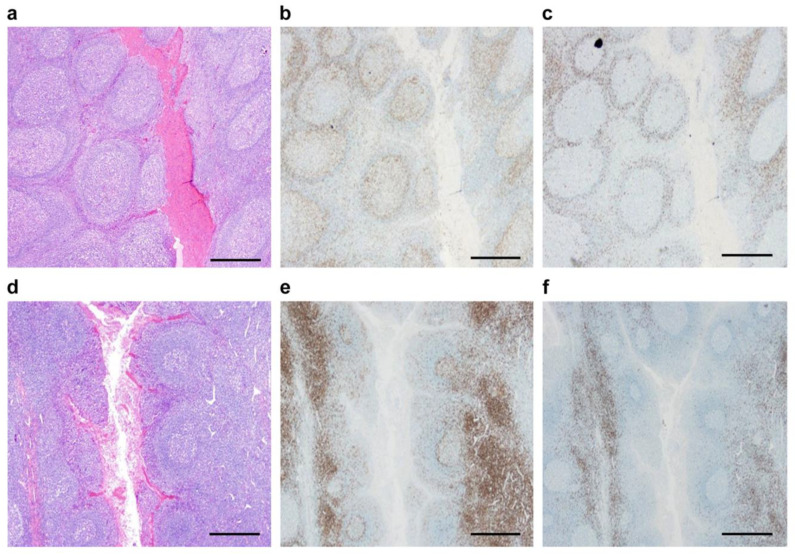
Histological and immunohistochemical comparison between TON-C (**a**–**c**) and TON-A (**d**–**f**). Hematoxylin–eosin stain (**a**,**d**). Anti-CD4 stain (**b**,**e**). Anti-CD8 stain (**c**,**f**). Scale bar: 500 μm.

**Figure 2 ijms-25-05298-f002:**
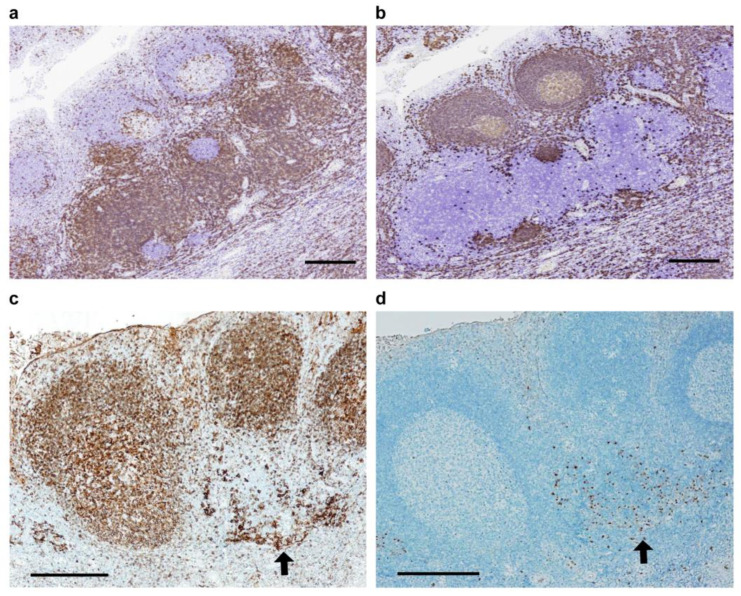
Immunohistochemical comparison between anti-CD3 antibody-positive T-lymphocytes (**a**) and anti-CD20 antibody-positive B-lymphocytes in TON-A (**b**). In the center of a CD3-positive T-nodule area, an assembly of HLA-DP, DQ, and DR-positive cells (arrow in (**c**)) revealed CD208-positive mature dendritic cells belonging to LAMP-3 (arrow in (**d**)). Scale bar: 500 μm.

**Figure 3 ijms-25-05298-f003:**
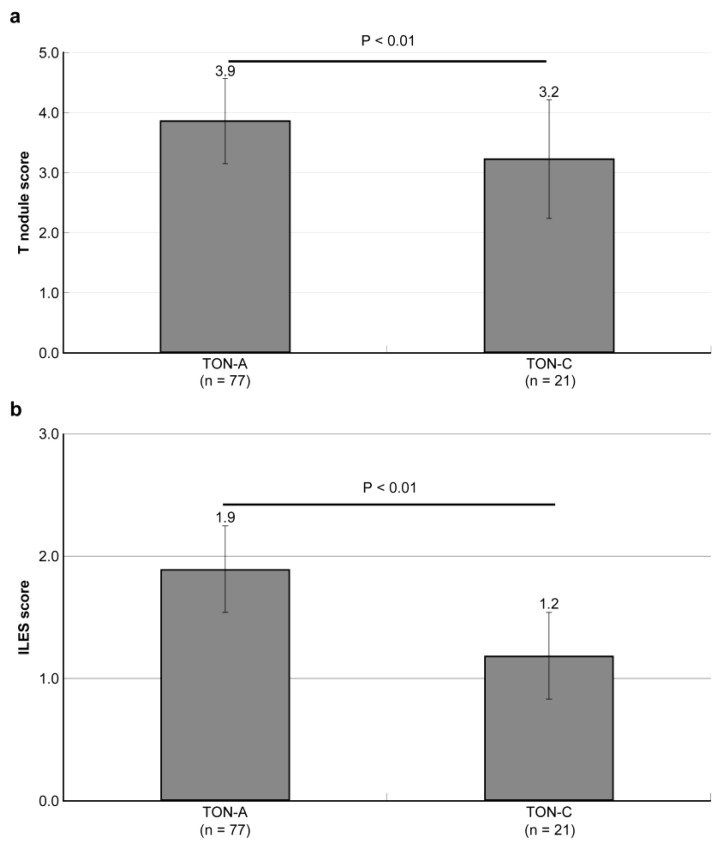
Statistical comparison of T-nodule and ILES scores between TON-A and TON-C. (**a**,**b**) Both T-nodule and ILES scores showed significantly higher values in TON-A compared to TON-C.

**Figure 4 ijms-25-05298-f004:**
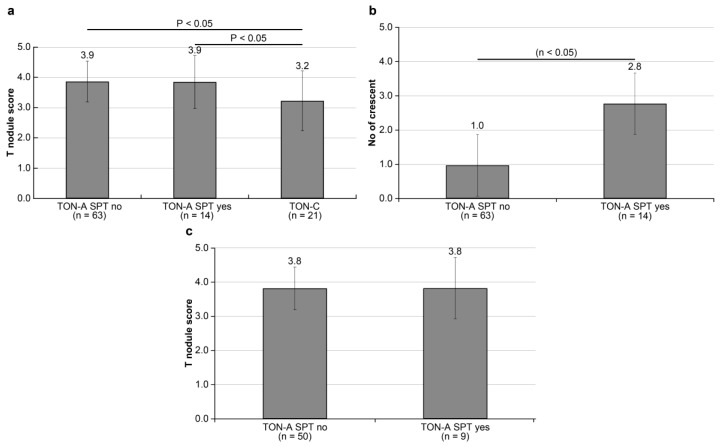
Statistical comparison of T-nodule score among SPT-preceding and tonsillectomy-preceding groups in TON-A and TON-C. The average T-nodule score demonstrated no difference between the tonsillectomy-preceding and SPT-preceding groups. The T-nodule scores in the TON-C group were significantly lower than those of tonsillectomy-preceding and SPT-preceding TON-A groups (**a**). The average number of crescents was significantly higher in the SPT-preceding group than that in the tonsillectomy-preceding group (**b**). However, no difference was found in the T-nodule scores between the tonsillectomy-preceding and SPT-preceding groups without crescents, respectively (**c**).

**Figure 5 ijms-25-05298-f005:**
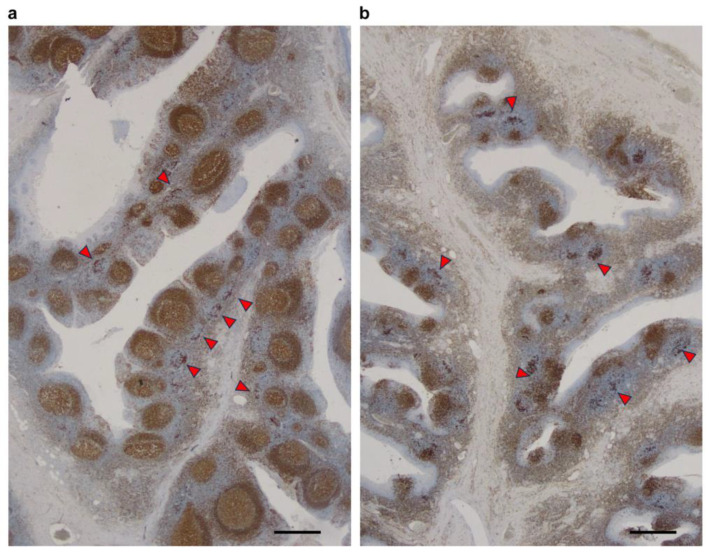
Immunohistochemical comparison between the SPT-preceding and tonsillectomy-preceding groups using anti-HLA- DP, DQ, and DR antibodies. The lymph follicles comprised of a germinal center and mantle zone were preserved with a marked number of T-nodules in the tonsillectomy-preceding TON-A group (arrowheads, (**a**)), whereas lymphatic follicles were severely destroyed, but the frequency of T-nodules was maintained in the SPT-preceding group (arrowheads, (**b**)). Scale bar: 500 μm.

**Figure 6 ijms-25-05298-f006:**
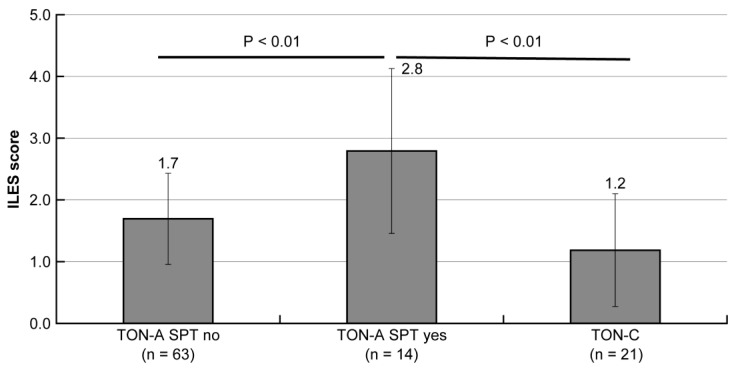
Statistical comparison concerning ILES score between the subgroups of TON-A and TON-C.

**Figure 7 ijms-25-05298-f007:**
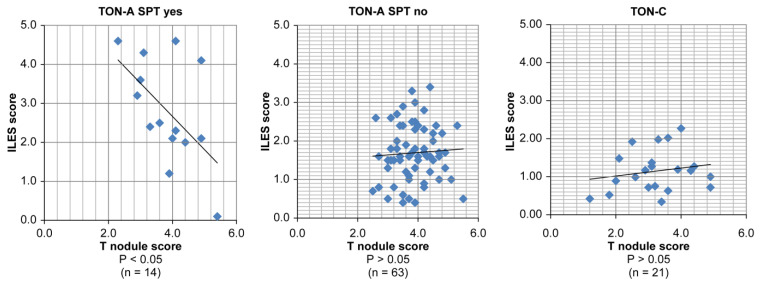
Statistical correlation between ILES and T-nodule scores in the subgroups of TON-A and TON-C.

**Figure 8 ijms-25-05298-f008:**
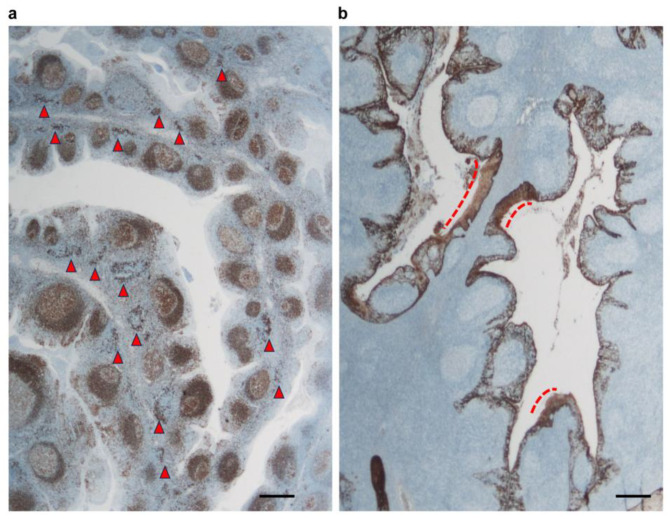
Method of quantification of the T-nodule score and ILES score. The number of cells that are positive for HLA-DP, DQ, and DR was counted in a 2.0 mm^2^ area and denoted as the T-nodule score (arrowheads, (**a**)). The percentage (%) of the length of the transformed keratotic epithelium out of the entire length of the crypt epithelium was estimated and showed 20% increments graded from 1 to 5 and expressed as the ILES score (dotted line, (**b**)). Scale bar: 500 μm.

**Table 1 ijms-25-05298-t001:** Correlation between T-nodule score in tonsil and renal lesions in patients with IgAN.

Variable	Probability	95% Confidential Interval	
		Lower Limit	Upper Limit
Mesangial hypercellularity	0.97	−0.01	0.01
Cellular crescent	0.02	0.01	0.15
Global sclerosis	0.12	0	0.03
Segmental sclerosis	0.11	−0.05	0
Fibrous crescent	0.43	−0.04	0.02
Interstitial fibrosis	0.87	−0.02	0.02

Multivariate linear regression analysis revealed that only the active crescent correlated significantly with the T-nodule (*p* < 0.05).

**Table 2 ijms-25-05298-t002:** Correlation between ILES score in tonsil and renal lesions in patients with IgAN.

Variable	Probability	95% Confidential Interval	
		Lower Limit	Upper Limit
Mesangial hypercellularity	0.14	0	0.02
Cellular crescent	0.75	−0.1	0.07
Global sclerosis	0.61	−0.02	0.03
Segmental sclerosis	0.02	0	0.07
Fibrous crescent	0.69	−0.03	0.04
Interstitial fibrosis	0.65	−0.03	0.02

Multivariate linear regression analysis revealed that only segmental sclerosis correlated significantly with the ILES scores (*p* < 0.05).

## Data Availability

Data are contained within the article and Appendix A. The raw data supporting the conclusions of this article will be made available by the authors on request.

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
