# Peer review of "Histological Correlation between Tonsillar and Glomerular Lesions in Patients with IgA Nephropathy Justifying Tonsillectomy: A Retrospective Cohort Study"

_ijms, 2024, doi:10.3390/ijms25105298_

Round 1

Reviewer 1 Report

Comments and Suggestions for Authors

Kensuke Joh et al. present an interest and original study, correlating histologies of tonsillar and glomerular lesions in IgA nephropathy patients. The manuscript is very well written, the study is adequately designed, the results are well presented and the discussion is comprehensively written.

Nevertheless, tonsillectomy does not represent standard therapy for IgA nephropathy in most parts of the world, while other novel therapies such as SGLT2i have shown to be efficient. I kindly ask the authors to add current standard treatment regimens to the Introduction section and mention that tonsillectomy is not internationally recommended. I would also appreciate a statement in the Discussion section, if authors would opt for tonsillectomy in all IgA NP cases and what the clinical consequence would be.

Otherwise, I support the acceptance of this well written manuscript.

Author Response

Thank you for your valuable comments.

We have modified the Introduction section and Discussion section as the reviewer 1 suggested (See red sentences in the revised manuscript.)

Reviewer 2 Report

Comments and Suggestions for Authors

Wonderful study and wonderful writing, and I recommend it for publication directly. 

Author Response

Thank you for your positive feedback

Reviewer 3 Report

Comments and Suggestions for Authors

This retrospective cohort study attempts to establish a correlation between tonsillar and glomerular lesions in patients with IgA nephropathy. There are several issues that need to be clarified before the manuscript can be considered for publication.

1. In general, images of immunohistochemical staining do no show specific positive signals. In Figure 2, CD4- and CD8-positive lymphocytes in T-nodules are not indicated; in Figure 3, it is difficult to see CD20-positive B-cells, HLA-DP/DQ/DR-positive cells, and CD208-positive mature dendritic cells. Better staining and higher magnification are required for these figures.

2. The authors may need to show histological or immunostained sections of renal tissues obtained from percutaneous needle biopsy, to illustrate glomerular lesions.

3. Scale bars are needed for all figures with tissue sections.

4. It is difficult to distinguish between main text and figure legends. If the paragraph under Figure 1 is a figure legend, it largely repeats the sentences in the main text.

5. The font size in Figure 5 is too small to be legible. The authors should try to use the same font size in different figures.

Author Response

  1. In general, images of immunohistochemical staining do no show specific positive signals. In Figure 2, CD4- and CD8-positive lymphocytes in T-nodules are not indicated; in Figure 3, it is difficult to see CD20-positive B-cells, HLA-DP/DQ/DR-positive cells, and CD208-positive mature dendritic cells. Better staining and higher magnification are required for these figures.

Our response

Thank you for your valuable comments. We have changed Figure 2 and Figure 3 by those with higher magnification.

  1. The authors may need to show histological or immunostained sections of renal tissues obtained from percutaneous needle biopsy, to illustrate glomerular lesions.

Our response

You have raised an important point; however, we believe that histological figures of glomerular lesions are not necessary in the manuscript because glomerular lesions such as crescent and segmental sclerosis are well known lesions but not particular lesions. There is no merit to occupy the space of histological figures of glomerular lesions in the manuscript.

  1. Scale bars are needed for all figures with tissue sections.

Our response

We have added Scale bars for all figures with histological sections.

  1. It is difficult to distinguish between main text and figure legends. If the paragraph under Figure 1 is a figure legend, it largely repeats the sentences in the main text.

Our response

Thank you for your comments. We agree with you. We have deleted the repeated sentences for Figure 1,2,3,4,5,6,7 and 8.

  1. The font size in Figure 5 is too small to be legible. The authors should try to use the same font size in different figures. 
  2. Our response

Thank you for your comments. We agree with you. We have changed the front size in Figure 5, which is similar to the other Figures.

Round 2

Reviewer 3 Report

Comments and Suggestions for Authors

The revise manuscript is improved. I have no further comments.